# Reverse water gas-shift reaction product driven dynamic activation of molybdenum nitride catalyst surface

Hui Xin[1,2,4], Rongtan Li ®[1,4], Le Lin[1], Rentao Mu[1], Mingrun Li[1], Dan Li ®[3] ✉, Qiang Fu ®[1] ✉ & Xinhe Bao ®[1] ✉

In heterogeneous catalysis catalyst activation is often observed during the reaction process, which is mostly attributed to the induction by reactants. In this work we report that surface structure of molybdenum nitride ($MoN_x$) catalyst exhibits a high dependency on the partial pressure or concentration of reaction products i.e., CO and $H_2O$ in reverse water gas-shift reaction (RWGS) ($CO_2:H_2 = 1:3$) but not reactants of $CO_2$ and $H_2$. Molybdenum oxide ($MoO_x$) overlayers formed by oxidation with $H_2O$ are observed at reaction pressure below 10 mbar or with low partial pressure of $CO/H_2O$ products, while CO-induced surface carbonization happens at reaction pressure above 100 mbar and with high partial pressure of $CO/H_2O$ products. The reaction products induce restructuring of $MoN_x$ surface into more active molybdenum carbide ($MoC_x$) to increase the reaction rate and make for higher partial pressure CO, which in turn promote further surface carbonization of $MoN_x$. We refer to this as the positive feedback between catalytic activity and catalyst activation in RWGS, which should be widely present in heterogeneous catalysis.

Designing advanced heterogeneous catalysts is of crucial importance for energy conversion and chemical production, which relies on controlling the surface active sites but still remains challenging[1–5]. It is generally recognized that each component of a reaction including reactant, product, and sometimes intermediates may interact with the catalyst, which can induce various structure changes[6–14]. Recently, competitive and/or synergistic effects of reaction components on catalyst structure have been observed, resulting in more complicated structure dynamics during reactions[15–21]. The restructuring of a catalyst usually leads to catalyst activation/deactivation[22–28] and subsequent change of local gaseous microenvironment surrounding the catalyst. In other words, the partial pressure or concentration of reactants and products near catalyst surface can be continuously varied by several orders of magnitude along with the catalyst surface restructuring during reaction process, i.e., higher partial pressure of the products for

activation but higher partial pressure of the reactants in case of deactivation. The fluctuating local gaseous microenvironment would drive the catalyst restructuring again, which means that both surface active structure and local gaseous microenvironment are changing during the ongoing reaction[15,16,29–33]. Therefore, a dynamic interplay between catalytic activity and catalyst structure may exist in any reactions, which is the key to understand the dynamics of surface active sites.

Here, we investigate dynamic response between catalytic activity and structural evolution of molybdenum nitride ($MoN_x$), an alternative to noble metal catalyst[34,35], in reverse water gas-shift (RWGS) reaction. To simulate the reaction microenvironment under the condition from very low to high conversion, the reaction pressure is controlled from 1 mbar to 10 bar to adjust the partial pressure of reaction components. The corresponding surface active structures at the above reaction

¹State Key Laboratory of Catalysis, Dalian Institute of Chemical Physics, iChEM, Chinese Academy of Sciences, Dalian 116023, China. ²Analytical & Testing Center, Sichuan University, Chengdu, Sichuan 610064, China. ³Key Laboratory of Green Chemistry and Technology, Ministry of Education, College of Chemistry, Sichuan University, Chengdu, Sichuan 610064, China. ⁴These authors contributed equally: Hui Xin, Rongtan Li. ✉e-mail: danli@scu.edu.cn; qfu@dicp.ac.cn; xhbao@dicp.ac.cn

pressures are investigated by using (quasi) in-situ X-ray photoelectron spectroscopy (XPS). We find that the reaction products of CO and $H_2O$ competitively dominate the surface structural evolution under different reaction pressures. $H_2O$-induced surface oxidation is prevailing at reaction pressure below 10 mbar. The effect of $H_2O$ and CO reaches balanced at 100 mbar making the surface almost keep at the pristine $MoN_x$ structure. When further increasing reaction pressure, CO-induced surface carbonization of $MoN_x$ to molybdenum carbide ($MoC_x$) is dominant at 1 bar and above. Sequentially, the surface carbonization of $MoN_x$ driven by CO enhances the reaction rate by two times compared with the fresh $MoN_x$. Once the reaction happens the formed reaction products around the catalyst surface drive the formation of active $MoC_x$ layers to enhance the catalytic activity and generate more reaction products, which displays a positive feedback between catalytic activity and evolution of active structure of $MoN_x$ catalyst during RWGS.

## Results and discussion

### Reaction pressure-dependent surface structure of $MoN_x$ catalysts

Tetragonal β-$Mo_2N$ catalyst was synthesized by a temperature-programmed nitridation process under $H_2/N_2$ (volume ratio of 3:1) mixed gases, followed by passivating under 1%$O_2$/Ar for 12 h at room temperature[36]. We then tested $CO_2$ hydrogenation performance over the as-prepared β-$Mo_2N$ catalyst at 500 °C and 1 bar (Supplementary Fig. 1), which shows high RWGS activity with $CO_2$ conversion of 43.1% and CO selectivity above 90.2%.

The surface structure in RWGS reaction has been investigated by (quasi) in-situ XPS in a near-ambient pressure XPS (NAP-XPS) system, which is composed of reaction and analysis chambers (Supplementary Fig. 2). Each catalyst is pre-reduced at 1 bar $H_2$ and 500 °C to remove most of surface passivated oxide layers. XPS N 1s/Mo 3p spectra of the pre-reduced β-$Mo_2N$ catalyst (treated at 1 bar $H_2$ and 500 °C) are

displayed in Fig. 1a, which show the main peaks at 394.1 and 397.8 eV belonging to $Mo^{\delta+}$ ($2 \leq \delta < 4$) and N atoms in β-$Mo_2N$ structure[37]. In addition, a few small components at 396.0 ~ 399.5 eV are from molybdenum oxide species ($MoO_x$, $2 \leq x \leq 3$) which can hardly be removed even after the reduction treatment[37–39] as confirmed by Mo 3d spectra in Supplementary Fig. 3 as well. A weak C 1s signal due to a small amount of residual contaminated carbon species also exists on the surface (Fig. 1a).

After exposing the pre-reduced catalyst to reaction gas at 1 mbar, the peaks belonging to $Mo^{\delta+}$ and N species in β-$Mo_2N$ almost disappear but those assigned to $MoO_x$ species become dominant in N 1s/Mo 3p and Mo 3d spectra (Fig. 1a and Supplementary Fig. 3) along with a strong O 1s signal, indicating occurrence of the severe surface oxidation. However, upon increasing the pressure to 10 mbar, 100 mbar and 1 bar, the O 1s signal gets weaker accompanied with the weakened $MoO_x$ peaks in N 1s/Mo 3p and Mo 3d spectra, which corresponds to less surface oxidation. Meanwhile, a new C 1s peak at 283.1 eV assigned to $MoC_x$ appears and gradually gets strengthened[40]. The N 1s peak for nitride N species reappears at 10 mbar and gets stronger at 100 mbar, which almost disappears again at 1 bar.

The evolution of various surface Mo species has been further illustrated by plotting atomic ratios of carbide C (from $MoC_x$), N (from $MoN_x$), and O (from $MoO_x$) relative to Mo as the function of reaction pressure (Fig. 1b). We observe the sudden increase in O/Mo but decrease in N/Mo with almost unchanged carbide C/Mo upon exposing the as-reduced β-$Mo_2N$ catalyst to 1 mbar reaction gas. The results indicate that surface N atoms coordinated with Mo atoms are substituted by O atoms under 1 mbar RWGS reaction gas. With increasing reaction pressure, the decreasing O/Mo but increasing carbide C/Mo have been observed while N/Mo starts to increase at 10 and 100 mbar and then decreases at 1 bar. As a result, $MoC_x$ species dominate the surface under 1 bar reaction gas while the maximum surface $MoN_x$ species exist under 100 mbar reaction gas. It can be speculated that the

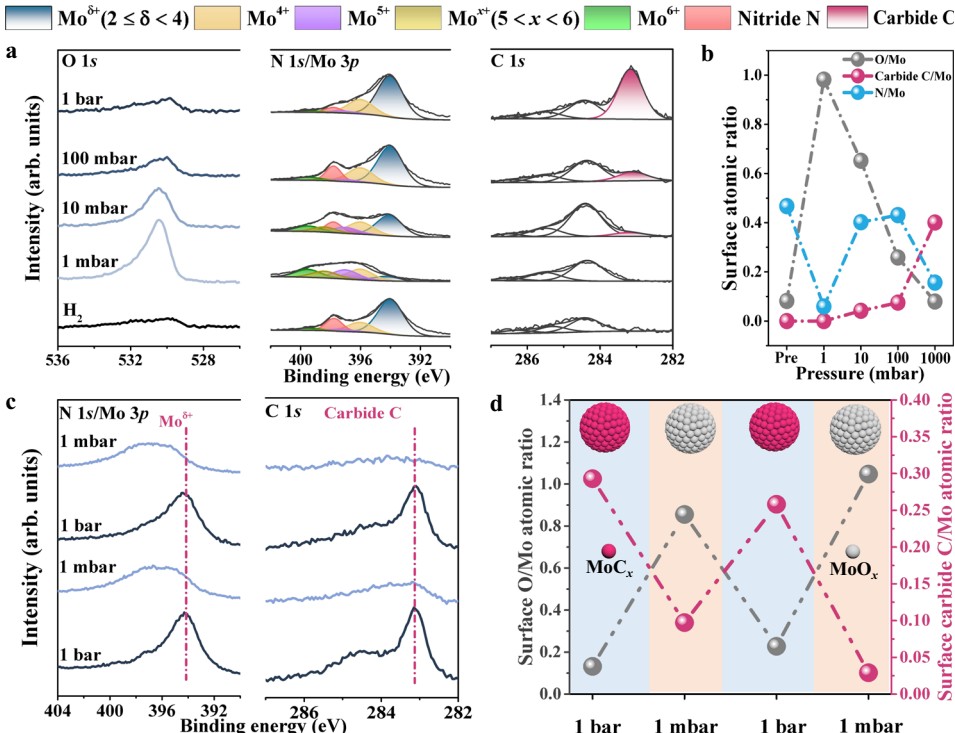

**Fig. 1 | Surface structure of β-$Mo_2N$ catalyst in RWGS reaction with different pressures at 500 °C as measured by quasi in-situ XPS. a** O 1s, N 1s/Mo 3p, and C 1s spectra acquired after reaction in 24%$CO_2$/72%$H_2$/$N_2$ from 1 mbar to 1 bar. **b** Surface atomic ratios of O, carbide C, and N relative to Mo as function of reaction pressure. **c** N 1s/Mo 3p and C 1s spectra acquired after reaction in alternative 1 bar and 1 mbar 24%$CO_2$/72%$H_2$/$N_2$ atmospheres (from bottom to top). **d** Surface O/Mo and carbide C/Mo atomic ratios calculated from C 1s, O 1s, and Mo 3d spectra in the cycled experiments with 1 mbar and 1 bar 24%$CO_2$/72%$H_2$/$N_2$ atmosphere.

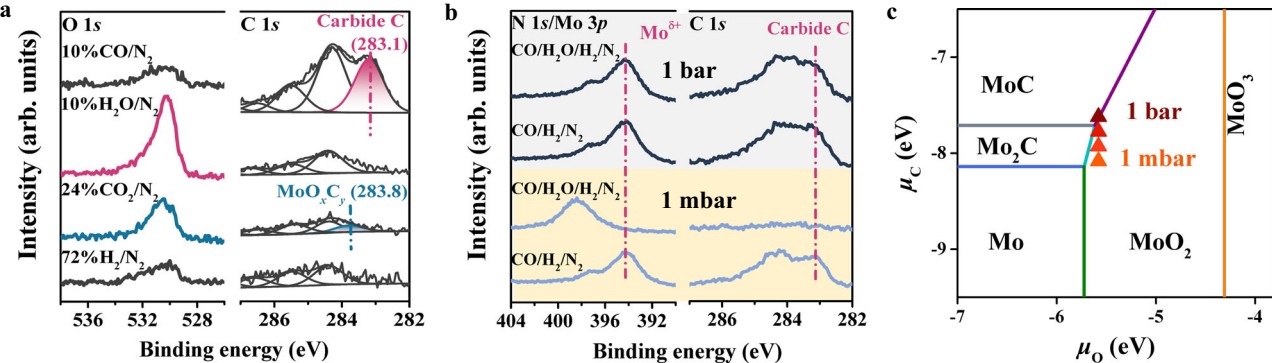

**Fig. 2 | Effect of each reaction component on surface structural evolution of β-Mo₂N. a** Quasi in-situ XPS O 1s and C 1s spectra of the pre-reduced β-Mo₂N exposed to 1 bar 72%H₂/N₂, 1 bar 24%CO₂/N₂, 1 bar 10%H₂O/N₂, and 1 bar 10%CO/N₂, respectively, at 500 °C for 30 min. **b** Quasi in-situ XPS N 1s/Mo 3p and C 1s spectra measured under 1 mbar (yellow region) and 1 bar (grey region) CO/H₂/N₂ and CO/H₂O/H₂/N₂ atmospheres, respectively. At each pressure, the pre-reduced β-Mo₂N was treated in CO/H₂/N₂ and then exposed in CO/H₂O/H₂/N₂. The ratio of H₂ is 62%; CO and H₂O in the mixed atmospheres are all 10%, similar to that in real reaction environment at 500 °C. **c** The computed phase diagram of Mo metal, carbides, and oxides. Each solid line denotes the boundary between two domains (phase equilibrium). The Mo species with the lowest free energy is marked for each domain. The triangles represent the position of $\mu_o$ and $\mu_c$ within specific reaction equilibrium conditions (details seen in Supplementary Table 1).

surface O atoms are removed and further replaced by C atoms, forming dominant surface $MoN_x$ and $MoC_x$ structures at 100 mbar and 1 bar, respectively. Mo atoms in $MoC_x$ exhibit similar binding energy of $Mo^{\delta+}$ species in β-Mo₂N[41], and thus Mo 3d spectra are nearly unchanged after the surface carbonization.

The β-Mo₂N catalyst has been subjected to the same treatment at 1 bar reaction gas but transferred in air for ex-situ XPS analysis. C 1s signal of carbides can be also observed but with much lower intensity after exposure to air for 5 min and totally disappears after exposure to air for 1 day (Supplementary Fig. 4), implying that the surface $MoC_x$ species can form in the ambient pressure reaction which are sensitive to the oxidizing components (O₂ and/or H₂O) in air[24,42,43] and form inert oxide overlayer on the surface. Besides, in-situ NAP-XPS experiments at 1 mbar have also been conducted, showing the similar results with quasi in-situ XPS results at 1 mbar, in which $MoO_x$ species dominates the surface (Supplementary Fig. 5). These imply that the effect of reaction pressure can't be ignored for the dynamic structural evolution.

We have shown that the surface structure of the nitride catalyst displays an interesting pressure dependence in the reaction gas. The surface has been further treated with 1 bar and 1 mbar reaction gases, alternatively. As shown in Fig. 1c and Supplementary Fig. 6, the surface dominated by $MoC_x$ after exposure to 1 bar reaction gas gets oxidized at 1 mbar reaction gas and the surface $MoO_x$ structure formed at 1 mbar is carbonized again at 1 bar reaction gas. Surface O/Mo and carbide C/Mo atomic ratios calculated from Mo 3d, O 1s, and C 1s (only the carbide species) peak areas reveal this reversible tendency more clearly (Fig. 1d). At 1 bar reaction gas, high carbide C/Mo but low O/Mo surface atomic ratios are observed, while this case gets reversed at 1 mbar pressure. Thus, the surface oxidation and carbonization processes can be well controlled by changing the reaction gas pressures between 1 mbar and 1 bar.

**Driving force for structural evolution of $MoN_x$**

To understand the structural change of the nitride surface in the reaction, the pre-reduced β-Mo₂N catalyst is exposed to each reaction component in RWGS i.e., 72%H₂/N₂, 24%CO₂/N₂, 10%H₂O/N₂, and 10% CO/N₂ at 1 bar and 500 °C for 30 minutes (the amount of CO and H₂O are calculated based on the CO₂ conversion results at 500 °C) and then quasi in-situ XPS measurements have been carried out (Fig. 2a). In 72% H₂/N₂ gas the catalyst surface keeps unchanged. In case of exposure to 10%H₂O/N₂, there are oxides on the surface as indicated by the appearance of a strong O 1s peak and a positive binding energy shift of

N 1s/Mo 3p signals (Supplementary Fig. 7). These manifest that product H₂O is more likely to oxidize the β-Mo₂N surface. When exposing the catalyst to 10%CO/N₂, a strong peak at 283.1 eV from carbides appears in C 1s spectra along with the weakened N signal in N 1s/Mo 3p spectra (Fig. 2a and Supplementary Fig. 7), indicating that CO can directly carbonize the β-Mo₂N surface. In addition, the peak around 284.4 eV is also significantly enhanced, suggesting the surface accumulation of graphitic carbon atoms from CO via the Boudouard process[44]. This surface reaction is similar to CO-induced formation of metal carbides on Fe-based catalysts during RWGS reaction[16,45]. Notably, if this CO-treated sample is further exposed to 72%H₂/N₂ the carbide peak intensity has been increased, indicating that H₂ can further promote the carbonization process (Supplementary Fig. 8). Upon exposing the β-Mo₂N catalyst to 24%CO₂/N₂, a weak peak at 283.8 eV from oxycarbide ($MoO_xC_y$)[40,46] appears in C 1s spectra and O 1s peak intensity has been slightly increased, implying that CO₂ can slightly carbonize and oxidize the surface. Since nitride catalysts can catalyze dissociation of CO₂ to CO (Supplementary Fig. 9)[47], the dissociated CO and O from CO₂ may cause the weak carbonization and oxidization, respectively.

The above results reveal that the reaction products H₂O and CO govern the surface structure of β-Mo₂N catalysts during reaction. To explore the effect of CO and H₂O gases at different partial pressures, we have investigated the surface structure of the pre-reduced β-Mo₂N catalyst exposed to CO/H₂/N₂ and CO/H₂O/H₂/N₂ under both 1 mbar and 1 bar conditions at 500 °C ensuring that the partial pressures of CO, H₂O and H₂ are the same as that in the corresponding reaction conditions (Fig. 2b). At 1 mbar CO/H₂/N₂ the surface structure is dominated by $MoC_x$ as confirmed by the obvious C 1s peak at 283.1 eV (Fig. 2b). From this surface, the exposure gas has been switched to 1 mbar CO/H₂O/H₂/N₂. Interestingly, the C 1s peak at 283.1 eV disappears and the Mo 3p peaks shift to higher binding energy positions, suggesting the severe surface oxidation, which is also verified by the appearance of a strong O 1s signal (Supplementary Fig. 10). The result manifests that the oxidation by H₂O surpasses the carbonization by CO at 1 mbar pressure. The surface carbides are also formed at 1 bar CO/H₂/N₂, which however remain unchanged with additional introduction of H₂O into the CO/H₂/N₂ gas (Fig. 2b), demonstrating that the carbonization by CO becomes dominant at 1 bar pressure no matter the coexistence with H₂O or not.

Apparently, both CO and H₂O competitively interact with β-Mo₂N, which are determined by varied oxygen and carbon chemical potentials at the different partial pressures. Figure 2c and Supplementary Fig. 11a display the phase diagram of bulk Mo-based compounds as a

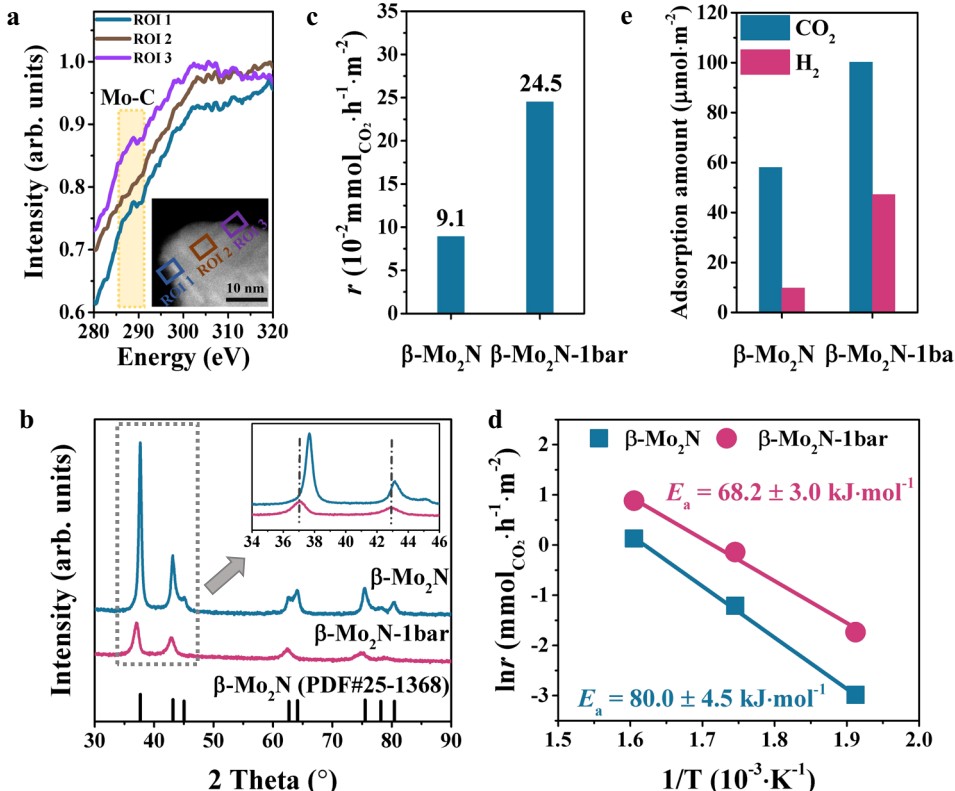

**Fig. 3 | Structure identification of β-Mo₂N catalyst after RWGS reaction at 1 bar and its corresponding catalytic activity. a** EELS spectra over the β-Mo₂N-1bar catalyst, which are recorded from regions indicated in the inset of HAADF-STEM image. **b** XRD patterns of the fresh β-Mo₂N and β-Mo₂N-1bar catalysts. **c** Reaction rate of $CO_2$ conversion on the β-Mo₂N catalysts normalized by the specific surface area under reaction condition of 50 mg catalysts, 250 °C, 24%$CO_2$/72%$H_2$/$N_2$, and WHSV = 30,000 mL·g$_{catal}^{-1}$·h$^{-1}$ with the $CO_2$ conversion below 5%. **d** Arrhenius plots of the fresh and β-Mo₂N-1bar catalysts with the $CO_2$ conversion below 16%, which are measured from 250 to 350 °C. The test of kinetic results has been done below 350 °C since the surface carbonization starts to happen above 350 °C (Supplementary Fig. 13), which could show the activity difference of the MoN$_x$ catalysts with and without surface carbonization. **e** The $CO_2$ and $H_2$ adsorption amount of fresh β-Mo₂N and β-Mo₂N-1bar catalysts calculated from $CO_2$/$H_2$-TPD experiments.

function of chemical potentials of oxygen ($\mu_o$) and carbon ($\mu_c$) obtained by density functional theory and ab initio thermodynamics calculations (see details in the Supplementary Information), which can be used to figure out the surface structure under the given conditions (triangles in the figure). Under all considered conditions H₂O possesses a higher oxidation ability (more positive $\mu_o$) than $CO_2$ (Supplementary Fig. 11b), which is consistent with the experiment results. When increasing the total reaction pressure but fixing ratios of all gas components, $\mu_o$ is unchanged but $\mu_c$ becomes more positive, which implies that $\mu_c$ may become more critical than $\mu_o$ in deciding the β-Mo₂N structure. Under low reaction gas pressure surface oxygen species are difficult to be removed due to low partial pressure of CO (lower $\mu_c$ case), while high partial pressure CO can consume surface oxygen species and leave carbon species behind favoring the carbonization of β-Mo₂N (higher $\mu_c$ case). Besides, the competition of CO and H₂O would reach a balance for CO in moderate partial pressure (middle $\mu_c$ case) such that few carbon or oxygen species are left on β-Mo₂N surface making β-Mo₂N intact (~100 mbar in Fig. 1a). As for this, we also exclude the possibility of nitridation by N₂ for the appearance of MoN$_x$, as seen from unchanged C 1$s$ and N 1$s$/Mo 3$p$ signals when the spent β-Mo₂N catalyst which treated at 1 bar reaction gas and 500 °C (denoted as β-Mo₂N-1bar) is exposed to 1 bar N₂ at 500 °C for 30 min (Supplementary Fig. 12). Overall, the thermodynamic scheme accompanied with experimental results show that increasing partial pressure of CO is in favor of carbonization of Mo₂N surface, well explaining the surface oxidation of β-Mo₂N below 10 mbar but the surface carbonization above 100 mbar.

## Dynamic activation of MoN$_x$ catalyst by reaction product-induced carbonization

The oxidation of β-Mo₂N catalyst in water gas-shift (CO + H₂O) and RWGS reactions has been revealed to occur in the surface region, forming nitride@oxide nanostructures with enhanced activity[48–50]. Here, we have further revealed that β-Mo₂N catalyst gets carbonized during RWGS reaction. In the following, structural characterization and catalytic performance test of the carbonized catalysts are performed. Scanning transmission electron microscopy coupled with electron energy loss spectroscopy (STEM-EELS) analysis of the β-Mo₂N-1bar reveals carbon K edges from the edge regions (region of interest (ROI) 1, 3) and center region (ROI 2) of a β-Mo₂N-1bar particle (Fig. 3a). At the edges, besides π* signal (about 285.4 eV) from graphite there is a peak at 288.6 eV ascribed to Mo-C σ* bond confirming the presence of MoC$_x$[51,52]. However, this signal at 288.6 eV is missing in the center region. The EELS results demonstrate that the MoC$_x$ species mainly form on the surface region.

X-ray diffraction (XRD) pattern of β-Mo₂N-1bar catalyst shows that its diffraction peaks shift to lower angle compared with the fresh sample, which is consistent with the larger lattice fringe (2.46 Å) than the fresh β-Mo₂N (2.38 Å) as observed in high-resolution TEM (HR-TEM) images (Fig. 3b and Supplementary Fig. 14). Thus, the structure of β-Mo₂N-1bar catalyst is suggested to be MoC$_x$ overlayers on β-Mo₂N with few C atoms doped in the β-Mo₂N bulk. Further increasing the reaction pressure to 10 bar, carbonization gets stronger and extends to the bulk as shown by the higher carbide C/Mo atomic ratio and the similar XRD pattern to α-MoC (Supplementary Fig. 15 and Fig. 16). This

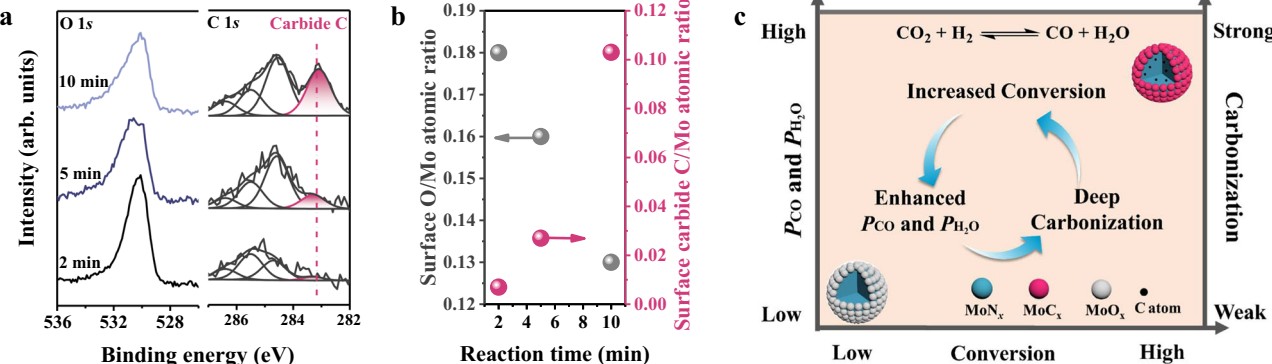

**Fig. 4 | Structural evolution of β-Mo₂N at 1 bar RWGS reaction atmosphere and 500 °C as measured by quasi in-situ XPS. a** O 1*s* and C 1*s* spectra acquired after reaction for 2, 5, 10 min, accordingly. **b** Surface O/Mo and carbide C/Mo atomic ratios calculated from C 1*s*, O 1*s*, and Mo 3*d* spectra from Fig. 4a. **c** Schematic diagram of the positive feedback between catalytic activity and structural evolution of β-Mo₂N under RWGS reaction.

phenomenon is in line with the formation of MoC phase at higher carbon potential from the phase diagram (Fig. 2c).

Interestingly, the β-Mo₂N−1bar catalyst shows higher RWGS activity (Supplementary Fig. 17). $CO_2$ conversion rate of this catalyst is above 2 and 3 times those of the fresh β-Mo₂N and α-MoC, i.e., 24.5 vs. 9.1 vs. 6.8 ($10^{-2}$ $mmol_{CO_2}·h^{-1}·m^{-2}$) under the identical reaction conditions (Fig. 3c and Supplementary Fig. 18) when the catalytic activity is normalized by their specific surface area (Supplementary Table 2, 17.1, 21.5, and 108.7 $m^2/g$ for the β-Mo₂N-1bar, fresh β-Mo₂N, and α-MoC accordingly). In addition, the apparent activation energy ($E_a$) for β-Mo₂N-1bar is 68.2 ± 3.0 kJ·mol⁻¹, which is lower than that of the fresh catalyst (80.0 ± 4.5 kJ·mol⁻¹) (Fig. 3d). As $CO_2$ and $H_2$ show negligible effect on the surface structure of β-Mo₂N (Fig. 2a), the $CO_2$/$H_2$-temperature programmed desorption ($CO_2$/$H_2$-TPD) experiments were conducted in order to illustrate surface adsorption and desorption of $CO_2$ and $H_2$. Figure 3e shows that the β-Mo₂N-1bar catalyst has higher adsorption amount for both $CO_2$ and $H_2$ than fresh β-Mo₂N. These manifest that the formed $MoC_x$ overlayers on Mo₂N surface improve the reactant adsorption, thus enhancing the catalytic activity. The CO-induced surface carbonization also occurs on a γ-Mo₂N catalyst during the same $CO_2$ hydrogenation reaction but with less enhancement of its catalytic activity compared with that of β-Mo₂N catalyst (Supplementary Figs. 19, 20). This is due to the lower carbonization degree of γ-Mo₂N in contrast with β-Mo₂N because of the weaker chemical adsorption of CO on γ-Mo₂N than that on β-Mo₂N (Supplementary Figs. 21, 22 and Table 3).

### The interplay between catalytic conversion and catalyst structure

To further reveal the structural evolution with the reaction activity, the surface structure of β-Mo₂N has been investigated in the kinetic region of RWGS reaction at 1 bar and in a closed reaction cell, in which the reaction time represents degree of the reaction conversion. As shown in Fig. 4a, a strong O 1*s* peak already appears at reaction time of 2 min but carbide C 1*s* peak at 283.1 eV is almost invisible, indicating the dominant surface oxidation process at the initial stage of the reaction. With the reaction time increasing from 2 to 5 and 10 min, the O 1*s* peak intensity gets weakened and carbide C 1*s* peak is strongly strengthened. The decreased surface O/Mo and increased carbide C/Mo atomic ratios displayed in Fig. 4b further confirm the transformation from oxidized to carbonized surface with the ongoing reaction at 1 bar.

The results suggest that the surface structure of $MoN_x$ dynamically responds to the local gaseous microenvironment with varying concentration of formed $H_2O$ and CO at different conversions. At the very beginning of the reaction, the low conversion with a small amount of $H_2O$ and CO products only induces the formation of surface $MoO_x$ (2 min), which is similar to the case at reaction pressure below 10 mbar. After prolonging the reaction time, conversion would increase generating more products. The enriched $H_2O$ and CO products facilitate the formation of surface $MoC_x$ (above 2 min), which is similar to the case at reaction pressure above 100 mbar. Therefore, it can be concluded that the reaction products ignite the surface restructuring (oxidation and carbonization) of $MoN_x$, enhancing the RWGS activity and increasing the partial pressure of CO near $MoN_x$ surface. The changing gaseous microenvironment in turn promotes the carbonization at the catalyst surface, which displays positive feedback between catalytic activity and structural evolution of $MoN_x$ in RWGS reaction as illustrated by Fig. 4c. Beyond the famous pressure gap mostly considering one specific reaction gas or the reactant[19], our findings demonstrate the importance of the response to products and catalytic activity in identifying the dynamics of active sites under catalytic conditions.

This work describes that the surface structure of $MoN_x$ catalysts is mainly determined by reaction products i.e., CO and $H_2O$ in the RWGS reaction. When the reaction pressure is below 10 mbar the partial pressure of CO/$H_2O$ is naturally low, such that $H_2O$ dominates the formation of Mo oxide ($MoO_x$) overlayer on $MoN_x$. With the reaction pressure above 100 mbar, the increased partial pressure of CO enables the surface carbonization of $MoN_x$, which suppresses the oxidization effect of $H_2O$ but produces Mo carbide ($MoC_x$) overlayers from CO. The generated $MoC_x$ overlayers enhances the reaction rate by two times compared with the pristine $MoN_x$. Therefore, the emergence of reaction products drives the surface restructuring of $MoN_x$ catalysts, enhancing activity and then increasing partial pressure of reaction products near $MoN_x$ surface, which can further strengthen the surface carbonization of $MoN_x$. Thus, positive feedback between catalytic activity and catalyst structure evolution during the reaction has been clearly demonstrated. This work highlights the dynamic interplay between local gaseous environment linked with catalytic activity and evolution of surface active structure during reactions.

## Methods

### Catalyst synthesis

Pure phase β-Mo₂N catalyst was synthesized using reported methods in literatures[36,53]. First, $MoO_3$ powder was obtained by calcination of ammonia heptamolybdate (($NH_4$)₆$Mo_7O_{24}$·$4H_2O$) at 500 °C for 4 h. After cooling down, the 0.5 g $MoO_3$ powder was transferred into a quartz tube and then nitridized under the flow of $H_2$ (100 mL/min)/$N_2$ (45 mL/min) mixed atmosphere to obtain β-Mo₂N. The detailed temperature programmed nitridation process was as follows: the temperature was linearly increased from ambient temperature to 300 °C with a heating rate of 5 °C·min⁻¹, and then the heating rate was

controlled at 1 °C·min⁻¹ from 300 to 700 °C maintaining at 700 °C for 2 h. After cooling down to room temperature, the obtained Mo nitrides catalysts were passivated in 1%O₂/Ar for 12 h to avoid violent oxidation upon exposure to air. The preparation method for γ-Mo₂N is identical to that of β-Mo₂N but using pure NH₃ (100 mL/min) as the nitridation atmosphere[37,53]. The α-MoC was obtained via carbonization of γ-Mo₂N under 20%CH₄/H₂ (100 mL/min) atmosphere[54]. The detailed temperature programmed carbonization process was as follows: the temperature was linearly increased from ambient temperature to 300 °C with a heating rate of 5 °C·min⁻¹, and then the heating rate was controlled at 1 °C·min⁻¹ from 300 to 700 °C maintaining at 700 °C for 2 h. After cooling down to room temperature, the obtained Mo carbide was passivated in 1% O₂/Ar for 12 h to avoid violent oxidation upon exposure to air.

## Catalytic tests

CO₂ hydrogenation reactions were tested using a homemade fixed-bed micro-reactor. The 50 mg pelleting catalysts (20 ~ 40 mesh) diluting with 200 mg SiC were loaded in a quartz tube with an inner diameter of 4 mm. The reactant gas consists of 24% CO₂, 72% H₂ (volume ratio), balanced with N₂, which was used as an internal standard. Before each measurement, the catalysts were pretreated by 75%H₂/N₂ at 500 °C for 2 h to remove the impurities adsorbed on catalyst surface, and then switched to reaction gas with the weight hourly space velocity (WHSV) of 30,000 mL·g$_{catal}$⁻¹·h⁻¹. The reaction started from 250 to 500 °C with interval of 50 °C at atmospheric pressure. Each temperature was maintained for 1 h. The effluent gas was online analyzed by Agilent GC6890N equipped with a TDX-1 column and thermal conductivity detector. The reaction kinetic parameters were obtained under low CO₂ conversion (<16%) by decreasing the reaction temperature.

## Catalyst characterization. X-ray diffraction (XRD)

The diffraction patterns of catalysts were performed using the Empyrean-100 diffractometer equipped with a Cu Kα radiation source (λ = 1.5418 Å) at 40 kV and 40 mA. The XRD patterns were collected with the 2θ value ranged from 30° to 90° at a scanning rate of 8°/min.

## Scanning Transmission Electron Microscopy (STEM)

HAADF-STEM image and electron energy-loss spectroscopy (EELS) elemental mapping were recorded on a JEM-ARM300F microscope operated at an accelerating voltage of 60 kV. The samples for TEM analysis were prepared by dropping ethanol suspension of samples on the carbon film-coated copper grids. High-resolution TEM (HRTEM) image was conducted on a JEM−2100 microscope operated at an accelerating voltage of 200 kV.

## Surface area of the catalysts

The textural properties of the samples were determined by N₂ adsorption on a Quadrasorb evo sorption analyzer at liquid nitrogen temperature (−196 °C). Prior to the measurements, the catalysts were treated under vacuum at 300 °C for 4 h. The specific surface area was determined by the Brunauer-Emmett-Teller (BET) method.

## X-ray Photoelectron Spectroscopy (XPS)

Quasi in-situ XPS experiments were carried out in a lab-based NAP-XPS system (SPECS EnviroESCA) with a monochromatic Al Kα x-ray source operated at 50 W. For quasi in-situ XPS experiments, the sample was treated in reaction chamber equipped in the NAP-XPS system with temperature from 250 to 500 °C at 1 bar or with pressures ranging from 1 mbar to 1 bar at 500 °C under a series of gas atmospheres (pure H₂, pure N₂, 24%CO₂/72%H₂/N₂, 24%CO₂/N₂, 72%H₂/N₂, 10%H₂O/N₂, 10%CO/N₂, 10%H₂O/62%H₂/N₂, 10%CO/62%H₂/N₂, 10%CO/10%H₂O/62%H₂/N₂). Then, the treated samples were cooled down to room temperature in the gas, evacuated in 1 minute, and transferred to the analysis chamber for XPS measurements. The order of pumping and

cooling doesn't affect the results (Supplementary Fig. 23). For another set of quasi in-situ XPS experiments at 1 bar and 10 bar, the sample after reaction was transferred from an air-tight reaction tube (micro-reactor for atmospheric-pressure or high-pressure reaction) to XPS analysis chamber without exposure to air with the aid of a glovebox and a mobile transfer chamber. The ex-*situ* XPS measurements for the sample taken from the micro-reactor were carried out on a spectrometer equipped with an Al Kα x-ray source operated at 300 W. All spectra were calibrated using the C 1s peak at 284.6 eV and all peaks were not normalized.

## Chemical adsorption/Desorption experiments

The temperature-programmed desorption experiments (CO₂/H₂/CO-TPD) were performed in a Micromeritics AutoChem II 2920. 0.1 g sample was put into the U tube. For the CO₂/H₂-TPD experiments, the fresh samples were pretreated in 75%H₂/N₂ at 500 °C for 1 h and then cooled to 50 °C. Then, switch the gas to 5%CO₂/N₂ or 10%H₂/N₂ for 1 h, and purge the samples for 30 min by helium. After that, the samples were heated from 50 to 800 °C at a rate of 10 °C·min⁻¹ under a flow of helium. For the spent samples, the fresh samples were pretreated in 75% H₂/N₂ at 500 °C for 1 h and then exposed to 24%CO₂/72%H₂/N₂ at 500 °C for 1 h. The subsequent steps were identical to that of the fresh samples. For the CO-TPD experiments, the steps were identical to CO₂/H₂-TPD experiments for the fresh samples, but use 3%CO/N₂ as the adsorbed gas. The CO₂/H₂/CO consumption amounts of catalysts were calculated by TCD signal, which was corrected by the standard Ag₂O sample.

## Temperature-programmed reaction experiment

The CO₂ temperature-programmed reaction (CO₂-TPRe) experiments have been performed on a Micromeritics Autochem 2920 chemisorption analyzer equipped with a Pfeiffer Vacuum OmniStar Mass Spectrometer (MS). Firstly, the sample was reduced under H₂ at 500 °C for 1 h to remove the surface oxide layer, and then introduced 5%CO₂/Ar. During the process, MS signals with various mass-to-charge (m/z) ratios were recorded to analyze the gaseous products.

## Theoretical calculations

All the theoretical calculations were carried out by using DFT and ab initio thermodynamics. Details of computational parameters, model constructions, and ab initio thermodynamics are presented in Supplementary Information. With assumption of equilibrium between the gas species under reaction condition, $\mu_o$ and $\mu_c$ were derived by the chemical potentials of the relevant gases, involving CO, CO₂, H₂, and H₂O.

H₂O shows much higher oxidation ability than CO₂ confirmed by XPS results in Fig. 2, and thus $\mu_o$ is mainly determined by H₂O, as shown in Eq. 1.

$$\mu_O = \mu_{H_2O} - \mu_{H_2} \tag{1}$$

For $\mu_c$ CO dominates the surface carbonization and thus $\mu_c$ is derived as Eq. 2.

$$\mu_C = \mu_{CO} - \mu_O \tag{2}$$

According to Eq. 3

$$\mu_{gas} = E_{gas} + ZPE + \delta H - TS + k_B T \ln \frac{P_{gas}}{P^\circ} \tag{3}$$

The chemical potentials of H₂O and H₂ are as below,

$$\mu_{H_2O} = E_{H_2O} + ZPE_{H_2O} + \delta H_{H_2O} - TS_{H_2O} + k_B T \ln \frac{P_{H_2O}}{P^\circ} \tag{4}$$

$$\mu_{H_2} = E_{H_2} + ZPE_{H_2} + \delta H_{H_2} - TS_{H_2} + k_B T \ln \frac{P_{H_2}}{P^\circ} \qquad (5)$$

Then, $\mu_O$ is

$$
\begin{aligned}
\mu_O &= \mu_{H_2O} - \mu_{H_2} \\
&= \left( E_{H_2O} + ZPE_{H_2O} + \delta H_{H_2O} - TS_{H_2O} + k_B T \ln \frac{P_{H_2O}}{P^\circ} \right) \\
&\quad - \left( E_{H_2} + ZPE_{H_2} + \delta H_{H_2} - TS_{H_2} + k_B T \ln \frac{P_{H_2}}{P^\circ} \right) \\
&= (E_{H_2O} + ZPE_{H_2O} + \delta H_{H_2O} - TS_{H_2O}) \\
&\quad - (E_{H_2} + ZPE_{H_2} + \delta H_{H_2} - TS_{H_2}) + k_B T \ln \frac{P_{H_2O}}{P_{H_2}}
\end{aligned}
\qquad (6)
$$

where $E_{gas}$ is the energy of gas phase species by DFT at 0 K. $ZPE$ is the zero point energy; $\delta H$ is the integral of heat capacity; $TS$ is the entropic temperature contribution; $k_B$ is the Boltzmann constant; $T$ is the absolute temperature; $P_{gas}$ is the partial pressure (Supplementary Table 1); and $P^\circ$ is the reference pressure (1 bar). At given $T$, the values of $ZPE$, $\delta H$, and $TS$ are constant and thus $\mu_O$ is just dependent on the ratio of $P_{H_2O}$ to $P_{H_2}$, which is constant. Thus, $\mu_O$ is unchanged with the total reaction pressure at fixing composition. For $\mu_c$ it is the dependent variable of the CO partial pressure at a fixed $\mu_O$ as defined by the Eqs. 2 and 3.

## Data availability

All data that support the findings in this paper are available within the article and its Supporting Information or are available from the corresponding authors upon request. Source data are provided with the paper.

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

## Acknowledgements

This research is supported by National Key R&D Program of China (2021YFA1502800, 2022YFA1504800 and 2022YFA1504500), National Natural Science Foundation of China (21825203, 22202198, and 22288201), Photon Science Center for Carbon Neutrality, and the Fundamental Research Funds for the Central Universities (20720220009). H.X. acknowledges the support from National Natural Science Foundation of China (22302135). R.L. thanks support from the China Postdoctoral Science Foundation (2023M743425).

## Author contributions

H.X. performed the catalyst preparation and catalytic reaction test. H.X. and R.L. performed the characterization and wrote the original draft. L.L. performed the DFT calculation. R.M. analyzed the data. M.L. carried out the TEM experiments. D.L., Q.F., and X.B. designed the study, analyzed the data, and revising the paper.

## Competing interests

The authors declare no competing interests.
