## [Peer Review File · Nature Communications]

Reverse water gas-shift reaction product driven dynamic activation of molybdenum nitride catalyst surfaceREVIEWER COMMENTS

Reviewer #1 (Remarks to the Author):

This is interesting and well-organized paper showing structural and chemical evolution of the MoN catalyst exposed to the CO₂ + H₂ reaction conditions at different pressures.

To the best of my knowledge, MoN as many other transition metal nitrides is not very promising for CO₂ hydrogenation reaction (see, for example, recent review doi.org/10.3389/fchem.2020.00452). Nonetheless, previous studies suggest that it is the carbide phase that is active in this reaction, which is in fact pure reverse-WGS reaction, although the CO₂: H₂ ratio 1:3 used here is typical for methanol synthesis. (As a side note, I think the authors should also specify this in the title since no other products were observed on their own catalysts). Therefore, for making a good catalyst for this reaction one needs to synthesize Mo carbide, using one or another precursor and carburization agent, with CO being the most obvious choice. Moreover, if one starts with MoN as in this study, then very first XPS experiments on the catalyst treated at 1 bar already show the formation of Mo carbide at surface, either as complete phase or an overlayer. The question how it happens and why it does not form at 1 or 10 mbar could be interesting, but not decisive for the final active catalyst. Actually, the results point to that typical NAP XPS studies at 1 mbar will be most likely not informative since quasi ex situ XPS results after 1-10 mbar would rather indicate Mo oxide as the active phase. (BTW, having such a setup in hand, why the authors have not done NAP XPS experiments for comparison?) Therefore, this study provides another example of the famous pressure gap. Although CO is a product and not a feed gas, it is CO₂ that is a source of carbon in this reaction, and the higher CO₂ pressure, the higher CO concentration, and hence the higher its chemical potential to accelerate carbide formation. Overall, there are no doubt interesting results presented in the manuscript, but not sufficiently important for Nat. Comm. In my opinion, regular catalysis-oriented journals would fit it better.

Minor:

I could not find information how exactly the sample was evacuated from the reactor. First cooling then pumping, or reverse, or simultaneously? Does the procedure affect the results? If the observed compositional changes are reversible (Fig. 1d), than cooling from 1 bar to vacuum that goes inevitably through the mbar range of pressures should transform carbide back to oxide. Apparently, kinetics is also important.

Figure 3 shows the reaction rate at 250°C (max 350°C) whereas all XPS studies were performed on samples reacted at 500°C. Why so?

Reviewer #2 (Remarks to the Author):

In this manuscript, the authors have characterized a MoN surface before and during CO₂ hydrogenation, as well as some probe reactions to support their findings. They clearly show dynamic evolution of the surface and a correlation to the changes in reactivity. Such results help push our field to better think about catalytic surfaces as non-static and how we should be careful in ex-situ measurements under non-reacting conditions. The findings are quite cool - Figure 1d is powerful in demonstrating the dynamics. And their time-based experiment towards the end was really meaningful also - these are quite nice findings.

Some things to make, in my opinion, the paper stronger:

1. I think a broader use of references would be appreciated.
2. Figure 1 - can the authors clarify if the peaks have been normalized (in panel a) or are they absolute and therefore intensities can be directly compared?
3. Lines 104 and 140 - where did the N atoms go?
4. I was surprised by the loss in C 1s peak with ambient exposure. This suggests that at room T reaction can occur? Can the authors expand on this?
5. Lines 176-179 - the authors put reactants in in a certain order. Would the same results hold if CO

was introduced first?

6. Line 243 - the TPD results are somewhat surprising - if doing a T ramp, should the CO₂ and H₂ not react to change the surface and therefore not truly represent adsorption and desorption? I think this needs added explanation.

Reviewer #3 (Remarks to the Author):

It is accepted that a working catalyst would restructure its surface at reacting atmosphere, however, to monitor such surface changes in situ in the reaction is quite difficult. In this contribution by Hui and coworkers, the evolution of surface species of MoN_x in (reverse) water gas shift reaction was systematically studied. Nitride, oxide and carbide were observed as a function of reactant pressure and composition. The findings were cross-proved by comprehensive (quasi) in situ characterizations. In addition, different surface area-normalized rates were attributed to the different surface species. In general, I highly recommend this work to be published. Prior to publication, I suggest authors to make the following changes.

1. In the title, "nitride" refers to a wide range of compound. But the manuscript studies only the MoN_x. So I suggest to make this more specific.
2. Figure 2c is the critical one for the explanation of oxide and carbide evolution based on thermodynamics. I do not quite get the idea "chemical potential of O is unchanged but chemical potential of C is increased, with the total reaction pressure at fixing composition". It would be better if a formula could be given in the main text showing how the chemical potential changes with pressure.
3. Figure 1b drives me to think of the source of N. As the pressure increased from 1 to 10 mbar, the nitride fraction increased, partly replacing oxide. What is the source of N? Does it mean that N₂ in the gas phase is not inert?
4. The formation of carbide reminds me of catalyst deactivation by cokes. Is any coke observed?

Point-by-Point Response to the Comments

Reviewer #1: This is interesting and well-organized paper showing structural and chemical evolution of the MoN catalyst exposed to the CO₂ + H₂ reaction conditions at different pressures.

To the best of my knowledge, MoN as many other transition metal nitrides is not very promising for CO₂ hydrogenation reaction (see, for example, recent review doi.org/10.3389/fchem.2020.00452). Nonetheless, previous studies suggest that it is the carbide phase that is active in this reaction, which is in fact pure reverse-WGS reaction, although the CO₂: H₂ ratio 1:3 used here is typical for methanol synthesis. (As a side note, I think the authors should also specify this in the title since no other products were observed on their own catalysts). Therefore, for making a good catalyst for this reaction one needs to synthesize Mo carbide, using one or another precursor and carburization agent, with CO being the most obvious choice. Moreover, if one starts with MoN as in this study, then very first XPS experiments on the catalyst treated at 1 bar already show the formation of Mo carbide at surface, either as complete phase or an overlayer. The question how it happens and why it does not form at 1 or 10 mbar could be interesting, but not decisive for the final active catalyst. Actually, the results point to that typical NAP XPS studies at 1 mbar will be most likely not informative since quasi ex situ XPS results after 1-10 mbar would rather indicate Mo oxide as the active phase. (BTW, having such a setup in hand, why the authors have not done NAP XPS experiments for comparison?) Therefore, this study provides another example of the famous pressure gap. Although CO is a product and not a feed gas, it is CO₂ that is a source of carbon in this reaction, and the higher CO₂ pressure, the higher CO concentration, and hence the higher its chemical potential to accelerate carbide formation. Overall, there are no doubt interesting results presented in the manuscript, but not sufficiently important for Nat. Comm. In my opinion, regular catalysis-oriented journals would fit it better.

Author reply: We thank the reviewer for the professional remarks and questions.

First of all, transition metal nitrides, like FeN_x, CoN_x, and Co₃Mo₃N catalysts, have been reported to display good catalytic activity to convert CO₂ into value-added

products including RWGS [*Angew. Chem. Int. Ed.* 2021, 60, 4496-4500; *Nat. Energy* 2017, 2, 869-876; *ACS Catal.* 2022, 12, 4696-4706], which motivates us to investigate the active structure of metal nitride catalysts in CO₂ hydrogenation reactions.

We agree with the referee that metal carbides are active for CO₂ hydrogenation reactions. The catalytic activity of synthesized α -MoC (the most studied carbides) has also been tested. Figure R1 shows the reaction data from both β -Mo₂N and α -MoC catalysts. When the catalytic activity is normalized by specific surface area (21.5 and 108.7 m²/g for the β -Mo₂N and α -MoC catalysts, respectively), the reaction rate of β -Mo₂N is higher than that of α -MoC i.e., 9.1 vs. 6.8 (10⁻² mmol_{CO₂}·h⁻¹·m⁻²). The results confirm the high activity of MoN_x catalysts in RWGS. The activity result of α -MoC has been added as Supplementary Fig. 18 in the revised Supplementary information with corresponding discussion on Page 10 in the revised manuscript: “CO₂ conversion rate of this catalyst is above 2 and 3 times those of the fresh β -Mo₂N and α -MoC, i.e., 24.5 vs. 9.1 vs. 6.8 (10⁻² mmol_{CO₂}·h⁻¹·m⁻²) under the identical reaction conditions (Fig. 3c and Supplementary Fig. 18) when the catalytic activity is normalized by their specific surface area (Supplementary Table 2, 17.1, 21.5, and 108.7 m²/g for the β -Mo₂N-1bar, fresh β -Mo₂N, and α -MoC accordingly).”

Figure R1. The catalytic activity results of fresh β -Mo₂N and α -MoC catalysts. (a) CO₂ conversion. (b) CO selectivity. Reaction condition: 50 mg catalyst, 1 bar 24% CO₂/72% H₂/N₂, WHSV = 30,000 mL·g_{catal}⁻¹·h⁻¹. (c) Reaction rate of CO₂ conversion on the β -Mo₂N and α -MoC catalysts normalized by the specific surface area under reaction condition of 50 mg catalysts, 250 °C, 24% CO₂/72% H₂/N₂, and WHSV = 30,000 mL·g_{catal}⁻¹·h⁻¹ with the CO₂ conversion below 10%.

We also agree with the referee that the reaction we studied is in fact the RWGS reaction. To make it clear, the manuscript title has been modified to “Reverse water gas-shift reaction product driven dynamic activation of molybdenum nitride catalyst surface” and other description for CO₂ hydrogenation reaction have also been modified in the revised manuscript.

As suggested by the referee, we conducted *in-situ* NAP-XPS experiment under 1 mbar reaction condition. Only MoO_x species are observed on the MoN_x surface (Fig. R2), which is similar to the quasi *in-situ* XPS after treatment at 1 mbar. The *in-situ* NAP-XPS results are added as Supplementary Fig. 5 in the revised Supplementary information with corresponding discussion on Page 5 in the revised manuscript: “Besides, *in-situ* NAP-XPS experiments at 1 mbar has also been conducted, showing the similar results with quasi *in-situ* XPS results at 1 mbar, in which MoO_x species dominates the surface (Supplementary Fig. 5). These imply that the effect of reaction pressure can’t be ignored for the dynamic structural evolution.”

Figure R2. *In-situ* NAP-XPS results of β -Mo₂N under 1 mbar 24% CO₂/72% H₂/N₂ at 500 °C showing the presence of surface MoO_x but not surface MoC_x species.

Moreover, it should be noted that in this work, MoN_x catalyst after typical reaction at 1 bar was delivered to XPS experiments, in which the air exposure time is less than 5 min. Some MoC_x species remained (Supplementary Fig. 4 in manuscript) and the short exposure time prevent the complete oxidation of surface formed MoC_x. If the used MoN_x catalyst was further exposed to air for 1 day (common *ex-situ* XPS experiments), no MoC_x species could be found (Fig. R3). Therefore, the surface MoO_x species rather than surface MoC_x species would be identified as the active sites of MoN_x catalyst based

on common *ex-situ* XPS and *in-situ* mbar NAP-XPS experiments. The above results also inspire us that the effect of reaction pressure can't be ignored for the dynamic structural evolution and was investigated in this work. The results are added into Supplementary Fig. 4 in the revised Supplementary information with corresponding discussion on Page 5 in the revised manuscript: "C 1s signal of carbides can be also observed but with much lower intensity after exposure to air for 5 minutes (Supplementary Fig. 4) and disappears after exposure to air for 1 day (Supplementary Fig. 4), implying that the surface MoC_x species can form in the ambient pressure reaction which are sensitive to the oxidizing components (O₂ and/or H₂O) in air^{42,43}, forming inert oxide overlayer on the surface."

Figure R3. The *ex-situ* C 1s XPS results of β -Mo₂N after reaction and subsequent exposure to air for 5 minutes and 1 day. The sample is taken from the micro-reactor after reaction under 1 bar 24% CO₂/72% H₂/N₂ at 500 °C for 1 h.

We appreciate very much the referee's comments on the gas pressure effect shown in our work. The most reported pressure gas effects are from one specific reaction gas or the reactant. In our work, we find that CO and H₂O products drive the surface evolution of MoN_x, which depends on their partial pressures. The surface structure may dynamically respond to the catalytic activity which changes the partial pressure of the products. The impact of this work is far beyond the pressure gap effect commonly discussed in literatures, and it reveals the dynamic response between catalytic activity and structural evolution of catalyst, which is important to understand the dynamics of

active sites during reaction but has not been well understood. We have modified the discussion on Page 12 in the revised manuscript to emphasize the importance and novelty of this work: “Beyond the famous pressure gap mostly considering one specific reaction gas or the reactant¹⁹, our findings demonstrate the importance of the response to products and reactants related to catalytic activity in identifying the dynamics of active sites under catalytic conditions.”

Minor:

1. I could not find information how exactly the sample was evacuated from the reactor. First cooling then pumping, or reverse, or simultaneously?

Author reply: We thank the referee for the question. Typically, the sample was cooled to RT in the reaction gases and then the reaction chamber was evacuated in 1 minute. Actually, we also reverse the procedure, first pumping to high vacuum and then cooling to RT. The order doesn't affect the results (Fig. R4). The results have been added as Supplementary Fig. S23 in the revised Supplementary information and corresponding discussion on Page 14 in the revised manuscript: “Then the treated samples were cooled down to RT in the gas and evacuated in 1 minute and transferred to the analysis chamber for XPS measurements. The order of pumping and cooling doesn't affect the results (Supplementary Fig. 23).”

Figure R4. Quasi *in-situ* XPS results of β -Mo₂N after treatment in 1 bar 24% CO₂/72% H₂/N₂ at 500 °C with different procedures. Cooling-Pumping: First

cooling in reaction gases and then pumping to high vacuum; Pumping-Cooling: First pumping to high vacuum and then cooling in vacuum.

2. Does the procedure affect the results? If the observed compositional changes are reversible (Fig. 1d), then cooling from 1 bar to vacuum that goes inevitably through the mbar range of pressures should transform carbide back to oxide. Apparently, kinetics is also important.

Author reply: We thank the referee for the critical question. Cooling from 1 bar to vacuum would go inevitably through the mbar range of pressures, but we don't observe the transformation from carbide back to oxide as displayed in Fig. R4. We agree with the referee that kinetics is important in this procedure. The evacuation from 1 bar to vacuum totally costs about 1 minute and the time for pressure at mbar is too short to complete the transformation. To make it clear, the results have been added as Supplementary Fig. S23 in the revised Supplementary information and corresponding discussion on Page 14 in the revised manuscript: “Then the treated samples were cooled down to RT in the gas and evacuated in 1 minute and transferred to the analysis chamber for XPS measurements. The order of pumping and cooling doesn't affect the results (Supplementary Fig. 23).”

3. Figure 3 shows the reaction rate at 250 °C (max 350°C) whereas all XPS studies were performed on samples reacted at 500°C. Why so?

Author reply: We thank the reviewer for raising this question. We have explored the surface state of MoN_x at different temperatures at 1 bar 24% CO₂/72% H₂/N₂. The results shown in Figure R5 indicate that the surface carbonization starts to happen above 350 °C. Thus, the activity comparison between MoN_x without surface carbonization and surface carbonized MoN_x should be tested at temperatures below 350 °C. The XPS results have been added as Supplementary Fig. 13 in the revised Supplementary information with corresponding discussion on Page 10 in the revised manuscript: “The test of kinetic results has been done below 350 °C since the surface carbonization starts to happen above 350 °C (Supplementary Fig. 13), which could show the activity

difference of the MoN_x with and without surface carbonization.”

Figure R5. C 1s XPS spectra of β -Mo₂N catalyst in CO₂ hydrogenation reaction with different temperatures at 1 bar 24% CO₂/72% H₂/N₂ as measured by quasi *in-situ* XPS.

Reviewer #2: In this manuscript, the authors have characterized a MoN surface before and during CO₂ hydrogenation, as well as some probe reactions to support their findings. They clearly show dynamic evolution of the surface and a correlation to the changes in reactivity. Such results help push our field to better think about catalytic surfaces as non-static and how we should be careful in ex-situ measurements under non-reacting conditions. The findings are quite cool - Figure 1d is powerful in demonstrating the dynamics. And their time-based experiment towards the end was really meaningful also - these are quite nice findings.

Author reply: We appreciate very much the reviewer’s positive remarks. It is a great encourage for us.

Some things to make, in my opinion, the paper stronger:

1. I think a broader use of references would be appreciated.

Author reply: We thank the reviewer for the nice suggestion. More relevant recent references have been added into the introduction section of manuscript. The newly cited references are refs. 4, 5, 14, 26, 27, 28, 42, and 43 in the revised manuscript.

2. Figure 1 - can the authors clarify if the peaks have been normalized (in panel a) or are they absolute and therefore intensities can be directly compared?

Author reply: Thank the reviewer for raising this question. The XPS peak intensities in Fig. 1 of manuscript are all absolute values, so they can be directly compared. We have clarified in the revised manuscript that the peaks are not normalized.

3. Lines 104 and 140 - where did the N atoms go?

Author reply: Thank the reviewer for the professional question. We speculate that surface N atoms are converted to NO_x species to desorb from the surface. We attempted to detect the possible formed NO_x species by mass spectrometer (MS). Unfortunately, no strong MS signal from NO_x species has been detected as shown in Figure R6 due to the small amount of surface oxidation products.

Figure R6. The MS result of $\beta\text{-Mo}_2\text{N}$ upon introducing 5% CO_2/Ar at 500 °C. The $\beta\text{-Mo}_2\text{N}$ is pre-reducing under H_2 at 500 °C for 1 h, and then introducing 5% CO_2/Ar .

4. I was surprised by the loss in C 1s peak with ambient exposure. This suggests that at room T reaction can occur? Can the authors expand on this?

Author reply: The loss in C 1s peak under the ambient exposure condition is mainly from the reaction of surface carbides. It is well known that the carbides are sensitive to air, which are easily oxidized, forming an inert oxide overlayer on the surface [*Science* 2017, 357, 389-393; *Nature* 2021, 589, 396-401]. The corresponding discussion has been added on Page 5 in the revised manuscript: “C 1s signal of carbides can be also observed but with much lower intensity after exposure to air for 5 minutes and

disappears after exposure to air for 1 day (Supplementary Fig. 4), implying that the surface MoC_x species can form in the ambient pressure reaction which is sensitive to oxidizing components (O₂ and/or H₂O) in air^{42,43} and forms inert oxide overlayer on the surface.”

5. Lines 176-179 - the authors put reactants in a certain order. Would the same results hold if CO was introduced first?

Author reply: Thank the reviewer for raising this question. Actually, at each pressure, CO was introduced first (mixed with H₂ and N₂, CO/H₂/N₂), and then the CO-treated sample was exposed to H₂O-containing gases (CO/H₂O/H₂/N₂). We have modified the description of the procedure on Pages 6, 7 in the revised manuscript to make it clear: “At each pressure, pre-reduced β-Mo₂N was treated in CO/H₂/N₂ and then exposed in CO/H₂O/H₂/N₂.”

6. Line 243 - the TPD results are somewhat surprising - if doing a T ramp, should the CO₂ and H₂ not react to change the surface and therefore not truly represent adsorption and desorption? I think this needs added explanation.

Author reply: Thank the referee for the constructive suggestion. Before activity test and all characterizations, the β-Mo₂N sample was pre-treated in H₂ at 500 and thus the effect of H₂ can be ignored. Fig. 2a in main text (Fig. R7 here) shows that CO₂ only has slight influence on the MoN_x surface structure even at 500 °C, thus we believe that the CO₂/H₂-TPD experiments could confirm the differences in adsorption performance for reactants over fresh β-Mo₂N and β-Mo₂N-1 bar catalysts. We have added corresponding discussion on Page 10 in the revised manuscript: “(As CO₂ and H₂ show negligible effect on the surface structure of β-Mo₂N (Fig. 2a), the CO₂/H₂-temperature programmed desorption (CO₂/H₂-TPD) experiments were conducted.)”

Figure R7. Quasi *in-situ* XPS O 1s and C 1s spectra of the pre-reduced β -Mo₂N exposed to 1 bar 72% H₂/N₂, 1 bar 24% CO₂/N₂, 1 bar 10% H₂O/N₂, and 1 bar 10% CO/N₂, respectively, at 500 °C for 30 minutes.

Reviewer #3: It is accepted that a working catalyst would restructure its surface at reacting atmosphere, however, to monitor such surface changes in situ in the reaction is quite difficult. In this contribution by Hui and coworkers, the evolution of surface species of MoN_x in (reverse) water gas shift reaction was systematically studied. Nitride, oxide and carbide were observed as a function of reactant pressure and composition. The findings were cross-proved by comprehensive (quasi) in situ characterizations. In addition, different surface area-normalized rates were attributed to the different surface species. In general, I highly recommend this work to be published. Prior to publication, I suggest authors to make the following changes.

Author reply: We appreciate very much for the reviewer's positive evaluation.

1. In the title, "nitride" refers to a wide range of compound. But the manuscript studies only the MoN_x. So I suggest to make this more specific.

Author reply: We thank the reviewer for the nice suggestion. According to the

suggestion, we have revised the manuscript title as “Reverse water gas-shift reaction product driven dynamic activation of molybdenum nitride catalyst surface”.

2. Figure 2c is the critical one for the explanation of oxide and carbide evolution based on thermodynamics. I do not quite get the idea “chemical potential of O is unchanged but chemical potential of C is increased, with the total reaction pressure at fixing composition”. It would be better if a formula could be given in the main text showing how the chemical potential changes with pressure.

Author reply: We thank the referee for the constructive comment. With assumption of equilibrium between the gas species under reaction condition, the chemical potentials of O (μ_o) and C (μ_c) were derived by the chemical potentials of the relevant gases, involving CO, CO₂, H₂, and H₂O.

H₂O shows much higher oxidation ability than CO₂ confirmed by XPS results in Figure 2 of manuscript, and thus μ_o is mainly determined by H₂O, as shown in Equation 1.

(Equation 1)

For μ_c CO dominates the surface carbonization, and thus μ_c is derived as Equation 2.

(Equation 2)

According to Equation 3

(Equation 3)

Chemical potentials of H₂O and H₂ are as below,

(Equation 4)

(Equation 5)

Then, μ_o is

where E_{gas} is the energy of gas phase species by DFT at 0 K. ZPE is the zero point energy; δH is the integral of heat capacity; TS is the entropic temperature contribution; k_B is the Boltzmann constant; T is the absolute temperature; P_{gas} is the partial pressure (Supplementary Table 1); and P^0 is the reference pressure (1 bar). At given T , the values of ZPE , δH , and TS are constant and thus μ_{O} is just dependent on the ratio of $P_{\text{H}_2\text{O}}$ to P_{H_2} , which is constant. Thus, μ_{O} is unchanged with the total reaction pressure at fixing composition. For μ_{C} , it is the dependent variable of the CO partial pressure at a fixed μ_{O} as defined by the equation 2 and 3. We have added formula in the revised manuscript on Pages 15, 16: “With assumption of equilibrium between the gas species under reaction condition, μ_{O} and μ_{C} were derived by the chemical potentials of the relevant gases, involving CO, CO₂, H₂, and H₂O.....”

3. Figure 1b drives me to think of the source of N. As the pressure increased from 1 to 10 mbar, the nitride fraction increased, partly replacing oxide. What is the source of N? Does it mean that N₂ in the gas phase is not inert?

Author reply: We thank the referee for the professional question. As shown in Fig. R8, the nitride cannot be recovered by exposing the β -Mo₂N-1bar to 1 bar N₂ at 500 °C. Thus, it's difficult for MoN_x or MoC_x to activate N₂ at lower N₂ pressure (0.04 - 0.4 mbar). As discussed in the manuscript, we believe that the effect of CO and H₂O reaches a balance at 10 to 100 mbar, which means that neither CO nor H₂O can dominate the surface and almost no oxygen and carbon atoms are left on the MoN_x surface, making the pristine nitride exposed. We have added more discussion on Page 8 in the revised manuscript: “the competition of CO and H₂O would reach a balance for CO in moderate partial pressure (middle μ_{C} case) such that few carbon or oxygen species are left on β -

Mo₂N surface making β -Mo₂N intact (~100 mbar in Fig. 1a). As for this, we also exclude the possibility of nitridation by N₂ for the appearance of MoN_x, as seen from unchanged C 1s and N 1s/Mo 3p signals when the spent β -Mo₂N catalyst which treated at 1 bar reaction gas and 500 °C (denoted as β -Mo₂N-1bar) is exposed to 1 bar N₂ at 500 °C for 30 minutes (Supplementary Fig. 12).”

Figure R8. Quasi *in-situ* C 1s and N 1s/Mo 3p spectra of the β -Mo₂N-1bar exposed to 1 bar N₂ at 500 °C for 30 minutes.

4. The formation of carbide reminds me of catalyst deactivation by cokes. Is any coke observed?

Author reply: Thank the reviewer for raising this question. C 1s XPS results of catalyst before and after reaction shown in Fig. 1a (Figure R9 here) display that only the carbide C peak is from a newly generated C species, and the other C species almost keep constant. This suggests that no coke forms during the reaction.

Figure R9. Quasi *in-situ* C 1s XPS spectra of β -Mo₂N catalyst in CO₂ hydrogenation reaction (24%CO₂/72%H₂/N₂) with different pressures at 500 ° C.

REVIEWERS' COMMENTS

Reviewer #1 (Remarks to the Author):

In their response, the authors have properly addressed all comments and questions raised by reviewers, and made all amendments requested. The manuscript can be accepted as it is.

Reviewer #2 (Remarks to the Author):

I think the authors have done a good job in addressing the concerns and again, interesting paper with such dynamics clearly shown.

Reviewer #3 (Remarks to the Author):

In the revised manuscript, authors have provided new experimental results and adequately addressed all my previous concerns. I think the current version can be published.

Reviewer #1: In their response, the authors have properly addressed all comments and questions raised by reviewers, and made all amendments requested. The manuscript can be accepted as it is.

Response: We really appreciate the referee for his/her nice suggestions which definitely help us to significantly improve the quality of our manuscript.

Reviewer #2: I think the authors have done a good job in addressing the concerns and again, interesting paper with such dynamics clearly shown.

Response: We thank the referee for the positive and constructive comments regarding our manuscript.

Reviewer #3: In the revised manuscript, authors have provided new experimental results and adequately addressed all my previous concerns. I think the current version can be published.

Response: We sincerely appreciate the referee for the valuable suggestions on our manuscript.